# OpenReview forum: "GuardAlign: Test-time Safety Alignment in Multimodal Large Language Models"
_ICLR.cc/2026/Conference — ICLR 2026 Poster_

### Official Review · Reviewer_Zbe8 · 2025-10-30

**Soundness:** 3
**Presentation:** 4
**Contribution:** 3
**Rating:** 8
**Confidence:** 3

**Summary:**

This paper focuses on safety alignment in multi-modal large language models (mllms). Recent studies reveal that input-side defenses detect unsafe images with CLIP and prepend safety prefixes to prompts, but most MLLMs suffer from inaccurate detection in complex scenes and unstable safety signals during decoding. To this end, the authors propose GuardAlign, which is a training-free defense framework that first leverages optimal transport to measure the distribution distances between image patches and unsafe semantics, then cross-modal attentive calibration ensures the safety signals remain consistently activated throughout generation. Extensive evaluations on six representative MLLMs demonstrate that GuardAlign reduces unsafe response rates by up to 39% on SPA-VL.

**Strengths:**

1.	This paper starts from an interesting motivation. In order to ensure the MLLMs are not guided by the dangerous information carried in the images, the authors come up with a method to mask the dangerous regions out of the images.

2.	The proposed method GuardAlign makes sense to me. To achieve the above goal, the authors propose OT-enhanced safety detection module, which first divides the image into patches, then measures the similarity between the patches and potential harmful content.

**Weaknesses:**

My main concern is about the negative effect that could be brought by the GuardAlign. Even though Table 6 demonstrates that their method preserves utility and yields consistent gains, the reasons behind are not revealed and discussed enough. Intuitively, masking the image leads to missing information about the image, but the performance is not affected. Is it because the masking region is small or something? If so, relevant analysis should be performed.

**Questions:**

Please refer to Weaknesses No.1.

---

> ### Author Response · Authors · 2025-11-20
> **Response to Reviewer Zbe8**
>
> Dear Reviewer Zbe8,
>
> Thank you for your thoughtful review and positive assessment of our work. Below, we address your comments and describe the corresponding revisions in the manuscript, with all updates highlighted in blue.
>
> ---
>
> The preservation of utility primarily stems from two factors in GuardAlign:
>
> 1.  OT-enhanced detection does not mask benign visual information. For harmless inputs, the OT cost remains above the threshold τ, so no patches are removed. As analyzed in Section 3.3 and Appendix B, OT effectively separates harmful and benign semantics, ensuring that informative regions are preserved.
>
> 2.  The attention calibration module helps the model retain visual cues that would otherwise fade during decoding. As shown in Figure 1, attention to image tokens decreases in deeper layers; our calibration module (Eq. 13) compensates for this drop, improving the model’s ability to make use of relevant visual information.
>
>
> These effects preserve useful visual evidence, which helps maintain utility and can lead to improvements.

---

### Official Review · Reviewer_vS6s · 2025-11-01

**Soundness:** 3
**Presentation:** 3
**Contribution:** 3
**Rating:** 6
**Confidence:** 4

**Summary:**

This paper proposes GuardAlign, a training-free defense framework for enhancing the safety of MLLMs. The method integrates two key strategies: OT-enhanced safety detection and cross-modal attentive calibration. Extensive evaluations on six MLLMs demonstrate that GuardAlign significantly reduces unsafe response rates while preserving the utility of MLLMs. The approach is efficient, adding minimal inference overhead compared to normal inference.

**Strengths:**

- Training-free efficiency: GuardAlign operates entirely at inference time without requiring additional data or fine-tuning, making it highly practical and resource-efficient.

- Comprehensive experimental validation: The paper provides thorough evaluations across multiple safety benchmarks and utility tasks, including detailed ablation studies and efficiency analyses.

- Low inference overhead: Compared to existing inference-time defenses like ETA, GuardAlign achieves better safety with moderate computational cost, striking a balance between effectiveness and efficiency.

**Weaknesses:**

- Utility improvement: While the paper reports that GuardAlign avoids performance degradation and even boosts utility (e.g., VQAv2 accuracy improves from 78.51% to 79.21%), the underlying mechanism is not sufficiently analyzed. It remains unclear why masking unsafe patches or calibrating attention would enhance general capabilities—this warrants further theoretical or empirical justification.

- Limited model scale evaluation: Experiments are confined to MLLMs up to 13B parameters (e.g., LLaVA-1.5-13B). Given the trend toward larger models (e.g., 70B+), validating GuardAlign on more scalable architectures would strengthen its generalizability and impact.

- Practicality concerns: The cross-modal attention calibration requires direct manipulation of attention maps within the MLLM backbone, which may involve accessing and altering internal model structures. This could limit applicability in black-box or proprietary systems where such modifications are restricted, reducing the method's versatility in real-world scenarios.

**Questions:**

Please see the weakness part.

---

> ### Author Response · Authors · 2025-11-20
> **Responese to Reviewer vS6s**
>
> Dear Reviewer vS6s,
>
> Thank you for your valuable and positive feedback.  Below, we provide responses to each of your comments and outline the modifications made to the manuscript. All revisions are highlighted in blue.
>
> ---
> **W1: Discussion on utility performance improvements.**
>
> A: The utility improvements mainly stem from the behavior of our two modules.
>
> 1. OT-enhanced detection does not suppress benign visual information: for harmless queries, the OT cost stays above the threshold τ, so no masking occurs. As discussed in Section 3.3 and Appendix B, OT reliably separates harmful from benign semantics, ensuring that informative regions are preserved.
> 2. As shown in Figure 1, attention to image tokens diminishes in deeper decoding layers; our calibration module (Eq. 13) re-amplifies these weakened image-token weights, allowing the model to better retain visual cues that would otherwise be overlooked.
>
> Overall, by preserving useful visual content and reinforcing the model’s reliance on it during decoding, GuardAlign yields the modest utility gains observed in our experiments. We have added more discussion in the revision.
>
> ---
>
> **W2: Evaluation on more large model.**
>
> A: To assess the scalability of GuardAlign, we further evaluate two larger LVLMs, InternVL2.5-38B and InternVL2.5-78B, following the evaluation protocol in [1]. GPT-4o scores model outputs on a 0–10 scale, where higher scores indicate safer responses. We adopt this scoring because USR and ASR become saturated on these models. As shown below, GuardAlign consistently improves safety at both scales:
>
> | Model | MM-SafetyBench(SD) | MM-SafetyBench(TYPO) | MM-SafetyBench(SD+TYPO) |
> |-------|:-------:|:----:|:--------:|
> | InternVL2.5-38B | 7.10 | 8.09 | 6.78 |
> | +GuardAlign | **7.45** | **8.51** | **7.23** |
> | InternVL2.5-78B | 7.43 | 7.22 | 6.60 |
> | +GuardAlign | **8.25** | **8.16** | **7.58** |
>
> These results show that GuardAlign continues to provide safety gains on 38B and 78B models, demonstrating good scalability and effectiveness on larger LVLMs.
>
> [1] VLMGuard-R1: Proactive Safety Alignment for VLMs via Reasoning-Driven Prompt Optimization, 2025, arXiv
>
> ---
>
> **W3: Applicability in black-box model.**
>
> A: Thank you for raising this concern. Our attention calibration module indeed operates in a white-box setting, but we believe this does not limit the contribution and scope of our work for the following reasons.
>
> 1. GuardAlign is a defense method. In both academia and industry, most safety defenses assume white-box access, since defense is typically deployed by the model provider rather than end users. In real deployment settings, model developers generally have full access to internal architectures, and white-box assumptions are standard for safety-critical mechanisms.
> 2. GuardAlign remains partially applicable in black-box environments. The OT-enhanced safety detection module functions independently of internal model structures and can be used even when the model backbone is inaccessible. Following recent black-box safety evaluation practices, we apply OT detection to Gemini-2.5-Flash and observe consistent safety improvements, as shown below:
>
> | Model | SPA-VL | MM-SafetyBench |
> |--------|:------:|:---------------:|
> | Gemini-2.5-Flash | 7.36 | 7.15 |
> | +GuardAlign (OT-ENHANCED DETECTION) | **7.41** | **7.28** |
>
> These results demonstrate that GuardAlign is not restricted to white-box usage and that its OT-based detection component remains effective even when applied to large proprietary black-box models. More results and analysis have been added to Appendix C.2(Lines 1026-1041).

---

### Official Review · Reviewer_xpBF · 2025-11-02

**Soundness:** 2
**Presentation:** 3
**Contribution:** 2
**Rating:** 4
**Confidence:** 4

**Summary:**

The paper proposes GuardAlign, a training-free framework to enhance the safety of multimodal large language models (MLLMs). It integrates two strategies: (1) OT-enhanced safety detection using optimal transport to accurately identify unsafe image regions, and (2) cross-modal attention calibration that reinforces safety prefix signals during text generation. Experiments on six major MLLMs demonstrate that GuardAlign significantly reduces unsafe response rates while maintaining or slightly improving task performance, achieving up to 39% reduction in unsafe outputs without retraining or additional data.

**Strengths:**

The paper is original in combining optimal transport-based detection with attention calibration for inference-time safety alignment.

The technical quality is solid, with rigorous theoretical analysis and comprehensive evaluations across models and datasets.

Clarity is high. both intuition and formulation are clearly articulated, and experimental design is systematic.

**Weaknesses:**

The method, while efficient, introduces several hyperparameters (e.g., τ, γ) that are not fully analyzed for stability or generalizability.

Evaluation is limited to vision–language reasoning; applicability to other modalities remains untested.

The detection component depends on CLIP backbones, which could inherit existing biases.

**Questions:**

How sensitive is GuardAlign's performance to its threshold and amplification parameters across unseen datasets?

Could the OT-based detection misclassify benign but semantically rich regions (false positives)?

Would combining GuardAlign with post-hoc fine-tuning methods yield additive benefits?

---

> ### Author Response · Authors · 2025-11-20
> **Response to Reviewer xpBF**
>
> Dear Reviewer xpBF,
>
> Thank you for your thoughtful review and constructive feedback! Below, we provide detailed responses to each of your comments and outline the modifications made to the manuscript. All revisions are highlighted in blue.
>
> ---
> **W1&Q1：Analysis of hyperparameters.**
>
> A： We have analyzed the threshold τ in Figure 4(a), showing that a moderate value consistently lowers USR with minimal impact on helpfulness, and retaining more patches further reduces USR while keeping utility unchanged. We also vary γ from 1.05 to 1.20 and report the results below. GuardAlign maintains stable performance across all tested values, with USR fluctuations within 0.2% and helpfulness changes within 0.2, demonstrating strong robustness to γ.
>
> | γ | Harm(USR)↓ | Helpful Score↑ |
> |---|:----------:|:--------------:|
> | **γ=1.05** | **10.63** | **8.26** |
> | **γ=1.10** | **10.42** | **8.47** |
> | **γ=1.15** | **10.33** | **8.56** |
> | **γ=1.20** | **10.31** | **8.63** |
> | **γ=1.25** | **10.35** | **8.58** |
>
> These results jointly indicate that GuardAlign is highly insensitive to its hyperparameters.
>
> ---
>
> **W2: Applicability to other modalities.**
>
> A： Our work is intentionally scoped to vision–language reasoning, where unsafe visual cues are most relevant. We agree that extending safety mechanisms to other modalities (e.g., audio–language, video–language) is important and leave this for future work. To address your concern, we made the following clarifications:
>
> 1. We replaced “multimodal large language models (MLLMs)” with “large vision–language models (LVLMs)” to accurately reflect the scope.
> 2. We explicitly state in the Introduction and Conclusion that GuardAlign is designed for vision–language settings.
> 3. We added a brief discussion in Section 6 LIMITATIONS & FUTURE DISCUSSION outlining how the framework might be adapted to other modalities and the associated technical challenges.
>
> ---
>
> **W3: Biases in CLIP backbone.**
>
> A: We appreciate the reviewer’s concern and acknowledge that CLIP encoders may inherit certain biases. However, our results indicate that these biases do not materially affect the effectiveness and conclusions of GuardAlign:
>
> 1. As shown in Figure 4 ( c ), we evaluate multiple visual encoders, including different CLIP variants and SigLIP. All backbones yield highly consistent trends and achieve comparable USR reduction, showing that GuardAlign is not sensitive to the specific characteristics or potential biases of a particular encoder.
> 2. CLIP-based models are widely adopted in vision–language safety, retrieval, and grounding tasks[1, 2, 3], where they consistently improve performance despite known biases. This broad empirical evidence suggests that residual backbone bias has a limited impact on the type of semantic-level detection performed in our framework.
>
> [1]Do We Really Need Curated Malicious Data for Safety Alignment in Multi-modal Large Language Models? 2025, CVPR
>
> [2]CLIP is Almost All You Need: Towards Parameter-Efficient Scene Text Retrieval without OCR, 2025 CVPR
>
> [3]Seeground: See and ground for zero-shot open-vocabulary 3d visual grounding, 2025 CVPR
>
> ---
>
> **Q2: Misclassification in OT-based detection.**
>
> A: Thank you for raising this point. Our OT-based detection can effectively distinguish benign but semantically rich regions from unsafe contents, as demonstrated in the following:
>
> 1. OT compares patches and unsafe semantics at the distribution level, enabling it to separate harmful semantics from benign yet visually complex regions. The theoretical analysis in Section 3.3 and Appendix B shows that OT assigns higher weights to discriminative textual variants, leading to stronger separation between unsafe patches and benign rich regions and thus reducing misclassification risk.
> 2. Figure 1(a) shows that OT-based detection provides a clear separation between harmful and benign patches. Building on this, we further evaluate the detection on VQAv2 and TextVQA datasets, which contain semantically rich but non-harmful regions, and observe no substantial tendency for OT to over-flag benign content.
>
> These results demonstrate that OT maintains stable behavior on benign yet semantically rich regions across both theoretical and empirical evaluations.
>
> ---

---

> > ### Author Response · Authors · 2025-11-20
> > **Response to Reviewer xpBF**
> >
> > **Q3: Combining GuardAlign with post-hoc fine-tuning methods.**
> >
> > A: GuardAlign is a training-free method and can be directly applied to any safety fine-tuned model. We adopt VLGuard [1] as the post-hoc SFT baseline and conduct additional evaluations. As shown below, SFT alone already improves safety (SPA-VL: 46.04 → 12.96), and applying GuardAlign on top of it further reduces unsafe responses (12.96 → 9.37) while preserving utility on TextVQA. More combined results and analysis have been added to Appendix C.2(Lines 993-1007).
> >
> > | Model | SPA-VL | TextVQA |
> > |-------|:------:|:-------:|
> > | LLaVA-1.5-7B | 46.04 | 58.20 |
> > | LLaVA-1.5-7B（Posthoc-SFT） | 12.96 | 57.80 |
> > | LLaVA-1.5-7B（Posthoc-SFT + GuardAlign） | **9.37** | **58.00** |
> >
> > [1] Safety fine-tuning at (almost) no cost: a baseline for vision large language models, ICML, 2024.

---

> ### Comment · Reviewer_xpBF · 2025-11-27
>
> Thanks for the update. My concerns are addressed

---

> > ### Author Response · Authors · 2025-11-28
> > **Heartfelt Gratitude to Reviewer xpBF**
> >
> > Dear Reviewer xpBF,
> >
> > Thank you for your kind feedback and for taking the time to review our updated work. We are grateful for your recognition and for increasing the rating—it means a lot to us and inspires us to continue improving.
> >
> > Your thoughtful comments helped us refine the analysis and present the GuardAlign framework more clearly, and we look forward to further strengthening its effectiveness in improving safety while preserving utility in vision-language models.
> >
> > Thank you again for your thoughtful comments and encouragement. We genuinely appreciate your support.
> >
> > Best regards,
> >
> > Authors

---

### Official Review · Reviewer_uARe · 2025-11-02

**Soundness:** 3
**Presentation:** 3
**Contribution:** 3
**Rating:** 4
**Confidence:** 3

**Summary:**

The paper proposes GuardAlign, a training-free, inference-time defense for multimodal LLMs (MLLMs) that combines two complementary modules:

OT-enhanced safety detection — use patch-level Optimal Transport (OT) between image patch features and a set of predefined “unsafe” prompt embeddings to identify and mask visual regions that align with harmful semantics;

Cross-modal attention calibration — amplify attention toward a prepended safety prefix across selected middle fusion layers so that the safety signal remains activated during autoregressive decoding.

**Strengths:**

+ GuardAlign operates at inference time and requires no fine-tuning, which is attractive for rapid deployment on large deployed MLLMs.
+ The coupling of fine-grained OT-based patch scoring with attention-level calibration for safety prefixes is an intuitive and novel pairing: detect & sanitize visual evidence, then ensure the LLM heeds the safety prefix.

**Weaknesses:**

- The method is evaluated on many benchmarks but primarily in a black-box or benchmarked adversary setting. An adaptive attacker that crafts images to both avoid OT detection and trigger unsafe generations (e.g., by distributing harmful signals over many patches or embedding signals in texture) is not evaluated. GuardAlign’s resilience to adaptive/strong adversaries is unclear.

**Questions:**

refer to weakness

---

> ### Author Response · Authors · 2025-11-20
> **Response to Reviewer uARe**
>
> Dear Reviewer uARe,
>
> Thank you for your constructive feedback and the positive assessment of our contribution and novelty.
> To address your concern, we evaluate two adversarial settings: attackers distributing harmful signals across patches, and attackers adapting their optimization in order to bypass the defense.
> We summarize the results and analysis below:
>
> | Method        | Pathc-PGD ε = 16/255 | Pathc-PGD ε = 32/255 | Auto-PGD ε = 16/255 | Auto-PGD ε = 32/255 |
> |--------------|:--------------------:|:--------------------:|:-------------------:|:-------------------:|
> | LLaVA-v1.5-7B| 69.09                | 74.55                | 80.60               | 90.30               |
> | ETA          | 5.78                 | 5.93                 | 4.27                | 4.48                |
> | **GuardAlign** | **2.15**           | **2.36**             | **3.23**            | **3.65**            |
>
>
>
> 1). GuardAlign remains robust even when harmful cues are intentionally distributed across multiple patches: ASR decreases from 69.09 → 2.15 (ε=16/255) and 74.55 → 2.36 (ε=32/255), showing that spreading harmful signals does not circumvent OT-based detection even under stronger perturbations, and in both cases GuardAlign also outperforms ETA.
>
> 2). Under an adaptive attacker that explicitly optimizes to defeat our defense, ASR is reduced from 80.60 → 3.23 (ε=16/255) and 90.30 → 3.65 (ε=32/255), demonstrating robustness against adaptive optimization. At ε=32/255, GuardAlign again surpasses ETA (4.48 → 3.65), indicating stable performance even under stronger adaptive attacks.
>
> These results and the analysis have been added to Appendix C.2 and are highlighted in blue (Lines 960–968,972-980).

---

### Author Response · Authors · 2025-12-03
**General Response**

Dear Area Chair,

We sincerely appreciate your time and evaluation, especially under a tight review schedule. We are glad that reviewers broadly recognized the **motivation, practicality, and technical soundness** of GuardAlign and highlighted key strengths of our framework.

---

## **Summary of Our Work and the Rebuttal Process**

* **Our Contributions:**
1. GuardAlign is motivated by the observation that unsafe behaviors in LVLMs are often triggered by **unsafe visual cues** and by **weakened safety signals** during decoding. This motivates an inference-time framework that filters harmful visual regions and reinforces safety guidance throughout generation.
2. Building on this perspective, GuardAlign combines **OT-based detection of unsafe image patches** with **lightweight cross-modal attention calibration**, enabling effective safety reinforcement **without any fine-tuning** while preserving the model’s utility.

*  **Rebuttal Summary**

The main reviewer concern centered on generalizability. After our detailed response, reviewer xpBF acknowledged the clarifications and indicated willingness to raise their score, improving the overall ratings from **[4, 4, 6, 8]** to **[4, 6, 6, 8]**.

---

## **Summary of Initial Reviews (Pre-Rebuttal Status)**

The reviewers recognized the following strengths during the initial review stage:

| Reviewer | Recognized Strengths |
|---------|----------------------|
| **uARe** | Highlighted GuardAlign's **training-free, inference-time design** and the **intuitive pairing** of OT-based patch scoring with attention-level safety calibration. |
| **xpBF** | Noted the **original integration** of OT detection and attention calibration, supported by **rigorous analysis** and **clear, comprehensive experiments**. |
| **vS6s** | Praised the **training-free efficiency**, **comprehensive evaluations** across safety and utility tasks, and the **moderate inference overhead**. |
| **Zbe8** | Appreciated the **strong safety motivation** (masking dangerous regions) and found the **OT-based patch detection** reasonable and effective. |


---

Reviewers primarily asked about:
(1) robustness to adaptive attacks,
(2) hyperparameter stability and OT behavior,
(3) scope, scalability, and black-box practicality,
(4) utility preservation,
(5) compatibility with safety fine-tuning.

Below is a concise summary of our revisions.

---

## **Summary of Main Concerns and Our Revisions**

| Comment | Modification in Revision | Location | Reviewer(s) |
|--------|--------------------------|----------|--------------|
| **Robustness under stronger/adaptive adversaries** | Added Patch-PGD and Auto-PGD; GuardAlign reduces ASR from **69–90% → 2–4%**, outperforming ETA. | `App. C.2` | **uARe** |
| **Hyperparameter sensitivity & OT behavior** | Swept τ and γ (≤0.2% variance); added theoretical and empirical evidence showing OT clearly separates harmful vs. benign regions. | `Sec. 3.3, Fig. 4, App. B, App. C.3` | **xpBF** |
| **CLIP backbone dependence** | Tested RN50, ViT-B/32, ViT-B/16, ViT-L/14, SigLIP; all show consistent improvements, indicating low sensitivity to encoder choice. | `Fig. 4(c), App. B` | `xpBF** |
| **Scope, scalability, black-box practicality** | Clarified LVLM scope; added InternVL2.5-38B/78B results with consistent gains; showed OT detection works for Gemini-2.5-Flash in black-box settings. | `Intro, Sec. 6, App. C.2` | **xpBF, vS6s** |
| **Utility preservation** | Explained that benign images rarely trigger masking; calibration restores weakened visual attention, preserving or slightly improving VQA utility. | `Fig. 1, Sec. 3.3, Tables 2 & 6` | **vS6s, Zbe8** |
| **Compatibility with SFT** | VLGuard SFT improves SPA-VL (**46.04 → 12.96**); GuardAlign further improves it to **9.37** while maintaining TextVQA. | `App. C.2` | **xpBF** |


---

## **Responses from Reviewers During Rebuttal**

After the rebuttal:
* Reviewer **xpBF** confirmed all concerns were addressed and **raised their score**.
* The remaining reviewers raised no new concerns.

| Reviewer ID | Original Rating | Response |
|-------------|-----------------|----------|
| **xpBF** | 4 → **6** | “All concerns addressed.” |
| **vS6s** | 6 | No further concerns. |
| **Zbe8** | 8 | No further concerns. |
| **uARe** | 4 | No further concerns. |

---

## **Our Expectation**

We sincerely appreciate the time and effort that the reviewers and the Area Chair have dedicated to evaluating our work, and we fully respect the objective and independent judgment involved in this process.

If any further clarification or additional material would be helpful, please feel free to let us know, and we will be glad to respond promptly.

With best regards,

Authors of Paper 8549

---

### Meta-Review · Area_Chair_43hZ · 2025-12-07

**Summary:**

The paper proposes a training-free safety framework, GuardAlign, which combines OT-based unsafe-patch detection with cross-modal attention calibration to stabilize safety signals during decoding.  In the initial round, reviewers generally found the idea intuitive and well motivated, but raised several concerns that placed the paper near the borderline. These included questions about robustness to adaptive attacks, the stability of OT thresholds and calibration hyperparameters, reliance on CLIP backbones, unclear explanations for the reported utility gains, limited testing on larger LVLMs, and concerns about white-box requirements for attention manipulation.  The authors provided a detailed and substantive rebuttal that included new experiments, theoretical clarifications, and expanded evaluations, which strengthened confidence in the overall contribution.

Overall, the AC recommends acceptance of the paper.

**Reviewer Concerns:**

The rebuttal successfully addressed most core issues, especially robustness under adaptive attacks, sensitivity to hyperparameters, utility preservation, scalability to larger models, and the role of the CLIP encoder. What remains less fully resolved are broader questions about applying GuardAlign beyond vision-language models and limitations inherent to its partial reliance on white-box access, which may restrict deployability in fully proprietary black-box environments. One reviewer acknowledged that the rebuttal had fully addressed their concerns.

**Reviewer Scores:**

xpBF: 6
Zbe8: 8
uARe: 4
vS6s: 6

---

### Decision · Program_Chairs · 2026-01-26

Accept (Poster)